# Heavy Metals in the Muscle and Hepatopancreas of Red Swamp Crayfish (*Procambarus clarkii*) in Campania (Italy)

**DOI:** 10.3390/ani11071933

**Published:** 2021-06-29

**Authors:** Andrea Ariano, Marcello Scivicco, Massimiliano D’Ambola, Salvatore Velotto, Rebecca Andreini, Simone Bertini, Annalisa Zaccaroni, Lorella Severino

**Affiliations:** 1Department of Veterinary Medicine and Animal Production, Division of Toxicology, University of Naples Federico II, Via Delpino 1, 80137 Naples, Italy; andrea.ariano@unina.it (A.A.); marcello.scivicco@unina.it (M.S.); m.dambola@hotmail.it (M.D.); lorella.severino@unina.it (L.S.); 2Department of Promotion of Human Sciences and the Quality of Life, University San Raffaele, Via di Val Cannuta 247, 00166 Roma, Italy; salvatore.velotto@uniroma5.it; 3Department of Veterinary Medical Sciences, University of Bologna, Viale Vespucci 2, 47042 Cesenatico, Italy; annalisa.zaccaroni@unibo.it; 4Department of Veterinary Science, University of Parma, 43126 Parma, Italy; simone.bertini@unipr.it

**Keywords:** heavy metals, crustaceans, bioindicator, anthropogenic pollutant

## Abstract

**Simple Summary:**

Heavy metals can represent a serious threat to marine and freshwater fauna through exposure, bioaccumulation and biomagnification processes. The aim of this study was to evaluate the presence of non-essential and essential elements in freshwater crayfish (*Procambarus clarkii*) edible tissues to establish the healthiness of this product and to evaluate the pollution status of the sampling sites from Campania region (Italy). The results suggest that crayfish were safe for human consumption and indicated mild contamination of heavy metals of the sampling areas.

**Abstract:**

The aim of this study was to carry out a quali-quantitative analysis of the presence of non-essential and essential trace elements in freshwater crayfish (*Procambarus clarkii*) edible tissues to establish the healthiness of this product and to evaluate the pollution status of the sampling sites included in the present study. *P. clarkii* is one of the most common species of freshwater crustaceans in Italy, regularly consumed by local people. Moreover, the crayfish, due to its trophic position and diet, can be considered as an excellent bioindicator of the health status of the ecosystem. We collected sixty crayfish samples from two different sites in Campania (Italy): Villa Literno and Sessa Aurunca. Concentrations of trace elements were determined by Inductively Coupled Plasma Optical Emission Spectroscopy (ICP-OES). Our data showed low concentrations of Cd, Hg and Pb, with values below the European Commission MRL (Commission Regulation (EC) 1881/2006). We suggest that data obtained from this study showed that crayfish collected from Villa Literno and Sessa Aurunca were safe for human consumption. Furthermore, the results of this research indicated mild contamination of heavy metals of the sampling sites, indicating a good health status of the area’s aquatic ecosystem.

## 1. Introduction

Trace elements are classified by the scientific community as non-essential and essential. Non-essential trace elements have no biological role in animal organisms and represent a serious threat to aquatic fauna. Heavy metals and metalloids such as arsenic, lead, cadmium and mercury originate from natural sources and human activities (mining, metal production, combustion of fossil fuels, sewage sludge and waste incineration) [1,2] and are spread worldwide in fresh and salty waters, becoming one of the major causes of persistent aquatic pollution. Trace elements enter the food chain through bioaccumulation and biomagnification processes, contributing to compromising the balance of the food chain for a long time [3]. Adverse effects linked to acute or chronic exposure to metals include damages to the immune system, helping the onset of infectious diseases, and interference with the endocrine system, leading to reproductive alterations. Among the freshwater fauna, crustaceans are one of the most sensitive macroinvertebrate species to suffer negative effects of exposure to metals due to their diet, way of feeding with direct contact with sediments, and life span [4,5,6], and they easily accumulate trace elements in the hepatopancreas, the target organ for metals investigation [7,8,9,10]. Since crustaceans are extremely sensitive to metal effects, are widely spread in aquatic ecosystems and are regularly consumed by humans, they represent an optimal bioindicator to gain information about the health status of the ecosystem and to determine safety and quality of food intended for human consumption. In our study, we focused on the red swamp crayfish *Procambarus clarkii* (Girard, 1852), which is common in the sampling areas we included in the study and usually consumed by local people. Moreover, *P. clarkii* is considered by the scientific community as an optimal bioindicator for trace elements contamination [11,12]. Indeed, the red swamp crayfish has been used as an indicator species to monitor the environmental quality and the contamination of biological habitats in previous studies [7,13,14,15,16,17]. Nowadays, the red swamp crayfish is listed in Italy as an invasive species. It originates from the United States and Mexico and arrived in Europe during the last century, for aquaculture purposes [18]. Unfortunately, most of the Italian farmers failed to take adequate precautions in their cultivation methods to prevent the crayfish escape from farm enclosers. Soon after, the red swamp crayfish established wild stable populations in many lakes and ponds across Italy and rapidly became the dominant freshwater crayfish [19,20]. Regarding the sampling areas, we focused our attention on geographic areas of the Campania region (Italy) which are well known to be characterized by high pollution of soil, fresh, salty water and groundwater. These sites represent ex-industrial areas and are located nearby illegal waste dumps [21]. Specifically, since the 1980s Naples and Caserta have been exploited as illegal landfills of toxic waste. Such operations and the accumulation of toxic products have had a serious impact on the ecosystem of the coast and the hinterland, influencing health and future development of the local fauna and human population [22]. In the present study, we performed a quali-quantitative analysis of trace elements in samples of hepatopancreas and abdominal muscle of *P. clarkii* collected in two different Italian sampling sites, selected for their potential high level of metal contamination. We sought to identify sources of pollution in the study area, to assess public health risk linked to consumption of crayfishes and to improve the current knowledge about the use of *P. clarkii* as a bioindicator of heavy metal pollution in freshwater ecosystems.

## 2. Materials and Methods

### 2.1. Sampling

Sixty samples of red swamp crayfish were collected during summer 2017. Crayfishes were captured using baited traps placed at Villa Literno (ViL), near the Volturno River, and at Sessa Aurunca (SeA), near the Garigliano River (Figure 1) in the Campania region. No data is at present available concerning pollution of the two areas, apart from one study reporting trace elements concentration in the blood of dogs from Sessa Aurunca [23]. Specimens were then transferred alive in refrigerated boxes (4–8 °C) to the laboratory. In our facility, crayfishes were weighed and sexed. Furthermore, we measured each carapace length using a caliper (Absolute Digimatic caliper, Mitutoyo, Japan) (Table 1), from the tip of the rostrum to the edge of the carapace. Crayfishes were euthanized by thermal shock (−80 °C for 30 min). Subsequently, the abdominal muscle and the hepatopancreas were removed under partially defrosting conditions and stored in Falcon tubes at −20 °C until further analyses.

### 2.2. Chemical and Instrumental Analysis

Each sample was homogenized and 0.5 ± 0.2 g of tissue was added to 5 mL of 65% HNO_3_ and 2.0 mL of 30% H_2_O_2_. Microwave-assisted digestion was performed with a specific mineralization program for 25 min at 190 °C. Samples were cooled at 32 °C and the digested mixture was transferred into a 50.0 mL flask and the final volume was obtained by adding Milli-Q water [24].

Trace elements detection and quantification were determined by ICP-OES technique using a Perkin Elmer Optima 2100 DV instrument coupled with a CETAC U5000AT. Subsequently, both metals quantification and quality assurance procedure were performed as described by Zaccaroni et al. [23]. LODs values (limit of detection values) as wet weight were: 0.024 µg g^−1^ for As; 0.0002 µg g^−1^ for Cu; 0.006 µg g^−1^ for Zn; 0.001 µg g^−1^ for Cr; 0.0018 µg g^−1^ for Cd; 0.011 µg g^−1^ for Pb; 0.001 µg g^−1^ for Hg. The performance of the method has been defined by interlaboratory studies organized by FAPAS (Food Analysis Performance Assessment Scheme, Sand Hutton, York, UK).

### 2.3. Statistical Analysis

Results are reported in wet weight as mean ± SEM (standard error) [25]. Statistical significance of the influence of sampling sites (ViL Vs. SeA) and statistical significance in concentrations of trace elements in target organs (muscle vs. hepatopancreas) were tested using factorial analysis of variance. Furthermore, we apply the ANOVA test to highlight differences between trace element accumulation in the hepatopancreas and the muscle and between the sampling areas. Multiple regression was used to discover statistical significance between trace element concentration and intrinsic variables (as total weight and gender of specimens). One-Sample Kolmogorov–Smirnov Test confirmed the normal distribution of our data. All our statistical analyses have been performed using MedCalc for Windows, version 18.11.3 (MedCalc Software, Ostend, Belgium). Significant value has been established at *p* < 0.05.

## 3. Results

Mean concentrations of As, Cu, Zn and Cr in abdominal muscle (AbM) and hepatopancreas (Hep) of *P. clarkii* are summarized in Figure 2.

Our results show a variability in the concentration of two trace elements in *P. clarkii*, depending on sampling sites. Specifically, the levels of As and Zn were significantly higher *(p* < 0.01) in *P. clarkii* tissue from ViL site. Significant differences in organ accumulation of As, Cr, Cu and Zn have been highlighted. Indeed, trace elements concentration was significantly higher in hepatopancreas than in muscle (Table 2.). In Hep, Arsenic was found at a mean concentration of 8.534 and 3.248 µg g^−1^, while in AbM mean values were 0.627 and 0.456 µg g^−1^, in ViL site (*p* < 0.01) and SeA site (*p* < 0.01), respectively. Our data show, both in samples from SeA site and Vil site, significant differences (*p* < 0.01) between Cu concentration in Hep and AbM. In addition, significant differences (*p* < 0.01) were found for Zn between Hep and AbM at ViL site. Finally, higher concentrations of Cr were found in the crayfish Hep compared AbM at both sampling sites (*p* < 0.01).

Results showed negligible levels of Cd and Pb in all samples of the crayfish AbM. In the Hep, Cd was found at a mean concentration of 0.020 and 0.018 µg g^−1^; Pb was found at a mean concentration of 0.015 and 0.012 µg g^−1^ in Villa Literno (ViL site) and Sessa Aurunca (SeA site), respectively. Mercury was found under the detection limit (dl) in all analyzed samples (Table 2).

The analyzed individuals varied in size and weight ranges, including males and females. The multiple regression analyses indicate that there were no correlations between weight, gender and concentration of all analyzed trace elements (*p* > 0.05).

## 4. Discussion

The absence of a relation between trace elements and gender agrees with other published studies on *P. clarkii* [13,16,26]. Moreover, we did not appreciate a significant link between trace element concentration in the analyzed tissues and the weight of specimens, suggesting that these parameters have a minor effect on metal accumulation in subjects inside the weight range considered in this study [24].

Arsenic concentrations found in the crayfish muscle are comparable to results obtained by Bellante et al. (0.537 µg g^−1^ w.w.). In the same study the concentration of As in hepatopancreas was lower than those found in the present study (1.128 µg g^−1^ w.w.) [16]. Comparable levels of As in muscle were found by Gedik et al. in crayfish from Lousiana [14]. Devesa et al. [27] report arsenic concentration ranging from 9.2 to 12 µg g^−1^ in muscle and from 2.5 to 2.6 µg g^−1^ in hepatopancreas of crayfish from Southern Spain, higher than those found in the present study. On the contrary, Mistri et al. [28] and Tan et al. [29] report mean As concentration in both Hep and AbM lower than those detected in present study.

Regarding essential trace element concentrations, previous studies reported variable values of copper and zinc levels in the crayfish tissues. Among them, Bellante et al. [16] reported Cu levels in crayfish hepatopancreas and muscle ranging from 1.149 to 48.3 µg g^−1^ (mean value 12.3 µg g^−1^) and from 1.34 to 12.72 µg g^−1^ (mean value 5.19 µg g^−1^) w.w., respectively. These data agree with the results of the present study. Similarly, Kuklina et al. [30] and Mistri et al. [28] report comparable Cu concentrations in both tissues. Despite this, another study conducted in Lousiana established a range for Cu and Zn concentrations in the crayfish muscle ranging from 23.8 to 44.2 µg g^−1^ and from 41.3 to 55.8 µg g^−1^, respectively [31]. Moreover, a recent study conducted in Central Italy showed Cu levels that varied from 23 to 1031 µg g^−1^ in Hep and from 27 to 187 µg g^−1^ in AbM [15]. Cu levels in the hepatopancreas and muscle reported by those authors were higher than those detected in the present study, while Zn levels in Hep and AbM were lower than those found in ViL and SeA sites.

Regarding Cr concentrations, Bellante et al. [16] reported levels in crayfish hepatopancreas and muscle of 0.915 µg g^−1^ and 0.24 µg g^−1^ w.w., respectively. Mancinelli et al. [13], reported Cr in muscle tissue of *P. clarkii* (0.20–0.29 µg g^−1^) at higher concentrations than those found in AbM in ViL and SeA. Kuklina et al. [30] and Tan et al. [29] report similar Cr concentrations in the Hep to those detected in present research, while levels detected in AbM are higher in Campania samples with respect to these two studies.

Detection of Cd and Pb has been widely explored in crayfish. The levels of Cd in AbM of ViL site and SeA site, respectively, are generally comparable with those found in the muscle of *P. clarkii* from Preola Lake (<dl–0.01 µg g^−1^ d.w.) and Gorgo Medio Lake (<dl–0.03 µg g^−1^ d.w.) in Sicily, Italy [16], and lower than those reported in crayfish muscle from Trasimeno Lake (0.05 µg g^−1^ and 2.2 µg g^−1^) and Bolsena Lake (0.03 µg g^−1^) in Central Italy [13,15]. The levels of Pb accumulated in AbM and Hep determined in our research are also lower than concentrations measured in other areas [15,16,17].

Cadmium concentrations found in Hep of ViL site and SeA site are comparable to those measured in hepatopancreas of *P. clarkii* from Preola Lake and Gorgo Medio Lake in Sicily, Italy [16], but lower than the ones reported by other authors [7]. In 2016, Goretti et al. [15], detected Cd (mean value 8.2 µg g^−1^ unpolluted area; 28.2 µg g^−1^ polluted area) and Pb (mean value 8.5 µg g^−1^ unpolluted area; 3.2 µg g^−1^ polluted area) in the hepatopancres of *P. clarkii* from Trasimeno Lake (Cental Italy) at higher levels than those found in ViL and SeA sites. Same results were reported for both Cd and Pb by Tan et al. [29], Mistri et al. [28] and Kuklina et al. [30].

The general evidence was that crayfishes from ViL and SeA accumulated higher levels of metals (As, Cu, Zn and Cr) in Hep than in AbM, in accordance with those reported in literature [7,14,15,16]. Almost all studies on the distribution of trace elements in crayfish tissues showed that the hepatopancreas is the target organ of storage and detoxification of heavy metals [7,8,9,10]. However, in the present study, no statistical differences were reported for Cd and Pb concentrations in AbM and Hep of *P. clarkii*, probably due to the negligible concentrations of these non-essential trace elements in the aquatic environment of both sampling sites.

### Concern for Public Health

Even though no European or Italian regulation for As, Cu, Zn and Cr concentration in crustaceans and food products exists (because they are considered as essential trace elements, necessary for specific physiological functions), some tolerable upper intake levels have been proposed by both American and EU governmental and research entities (National Institutes of Health, U.S., Department of Health and Human Services; German Federal Institute for Risk Assessment, BfR; Scientific Committee on Food, European Commission, SCF; EFSA).

Copper is easily found in the environment and is essential for normal growth and metabolism [32]. Additionally, it is a component of the respiratory metalloprotein hemocyanin in crustaceans [33]. Therefore, relatively high copper amounts may be found in crayfish tissues, mainly in the hepatopancreas [7,34]. The role of Cu in crayfish metabolism and its great variability in data reported by other studies make comparison difficult, but the concentrations of Cu found in the present study are generally similar or higher than those reported in nine crayfishes captured both in polluted and unpolluted study areas [16]. Detected levels of copper are well above the recommended dietary allowances for toddlers and for adults (0.14–0.15 µg g^−1^ respectively) set by NIH [35] and of 0.08 µg g^−1^ for adults defined by BfR, SCF and EFSA [36,37,38].

The concentrations of Zn were higher than concentrations found by other authors in polluted and unpolluted areas [16,31,39]. Our results are indicative of high Zn levels, especially in the ViL site. These levels exceed the tolerable upper intake level (UL) defined by the SCF of 0.41 µg g^−1^ for adults [40].

Anyway, it should be noted that crayfish consumption is not that common among the Italian population, and the quantity of flesh usually consumed is generally reduced, so for both Cu and Zn a reduced exposure, and consequent health risk, is expected.

Chromium levels detected in Hep and AbM are comparable or lower than those reported by other authors [13,16]. Furthermore, Cr concentrations in AbM are below the threshold concentration suggested by FDA [41] of 1.089 µg g^−1^ w.w. for human consumption. Anyway, it is important to remember that also an excess of these metals can potentially cause harmful effects in organisms [10,42]. No UL has been defined for Chromium, but the WHO suggested to not exceed a 250 µg/day supplementation, equivalent to a daily dose of 4.16 µg g^−1^, if using a standard weight of 60 kg [43,44].

Although our results are suggestive of higher levels of As in Hep, especially in the ViL site, concentrations of As in AbM are comparable to those reported in the literature and considered concentration responsible for low risk for human consumption [14]. No UL has been set for As at present by any governmental institution, but a maximum concentration of 50 µg L^−1^ has been defined [35], well below the mean concentrations detected in present study. Anyway, it should be remembered that the substantial portion of arsenic present in fish and mollusks is in the organic form and, as stated by Trumbo et al. [35] as well, these forms are less toxic than inorganic form (for whom the assessment is done). Consequently, any increased health risk from food products such as fish and mollusks is unlikely.

Regarding non-essential trace elements, The European Union legislation (Commission Regulation (EC) 1881/2006 and its amendment (Commission Regulation (EU) 420/2011) on food safety clearly establish the MRLs for total Cd, Pb and Hg which can be detected in the muscle of crustaceans (0.5 µg g^−1^ w.w. for Cd; 0.5 µg g^−1^ w.w. for Pb and 0.5 µg g^−1^ w.w. for Hg) intended for human consumption [45,46]. The results obtained in the current study show lower levels of Cd, Pb and Hg in AbM and Hep from ViL and SeA sites than the MRLs reported by EU regulations. Furthermore, our data are largely below the established MRLs, suggesting a limited Cd, Pb and Hg contamination of the aquatic environment of the study areas, and good food safety of aquatic products derived from these geographic areas.

## 5. Conclusions

The accumulation of trace elements in *P. clarkii* tissues reflects the concentrations of metals in the surrounding environment [5] and our data suggest that *P. clarkii* could be considered a good bioindicator for metal pollution. The higher Cu and Zn concentrations found in *P. clarkii* tissues, especially for Zn from ViL site, could be related to higher anthropic activity in these areas, as already proved by a paper by Zaccaroni et al. [23]. However, these results must be evaluated with caution because of the small number of samples collected and the lack of legal limits for the detection of some trace elements concentration in crustaceans and other fish products. The higher As concentrations in crayfish Hep, especially from ViL site, must be further clarified in order to identify possible sources of contamination in these areas. Further studies are also needed in determining the percentage of organic and inorganic arsenic in crayfish tissues.

Ongoing studies on metals in a greater number of *P. clarkii*, in other biological and environmental samples and in other geographical areas, will provide more useful information to confirm this species as indicator of environmental contamination.

## Figures and Tables

**Figure 1 animals-11-01933-f001:**
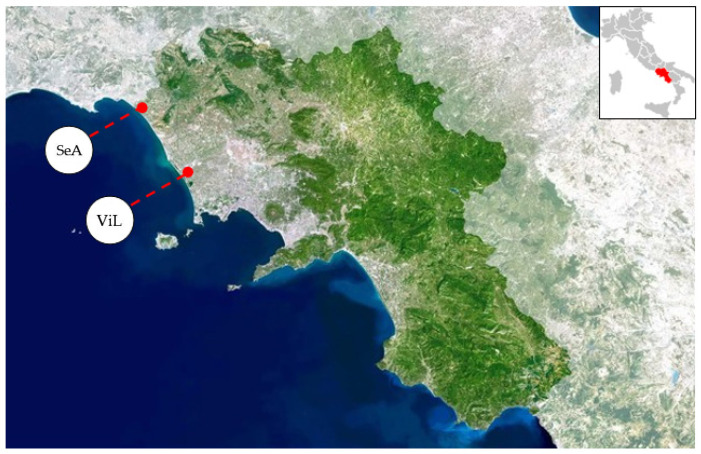
Map showing locations of the sampling sites: Villa Literno (ViL) and Sessa Aurunca (SeA).

**Figure 2 animals-11-01933-f002:**
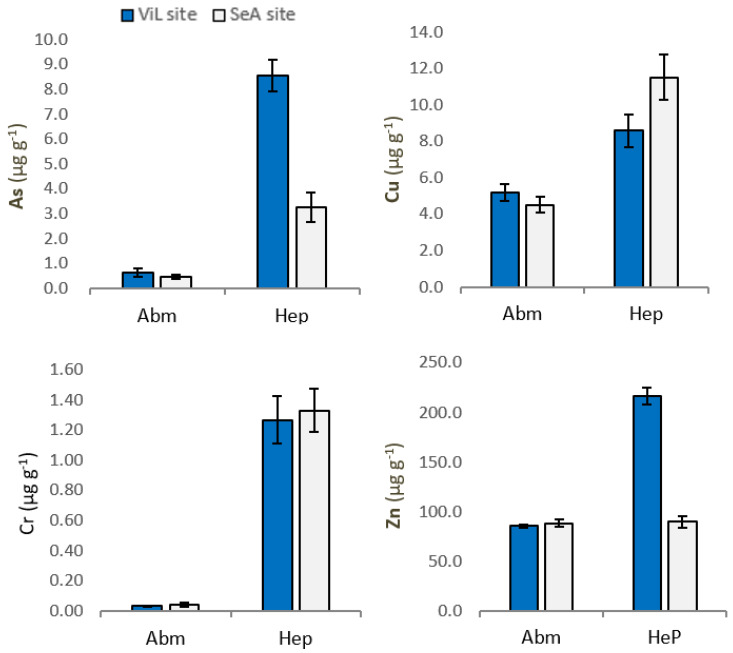
Concentrations of As, Cu, Zn and Cr in Procambarus clarkii abdominal muscle (AbM) and hepatopancreas (Hep) from Villa Literno (ViL) and Sessa Aurunca (SeA). Vertical bars represent average concentration (µg g^−1^ wet weigth) ± SEM.

**Table 1 animals-11-01933-t001:** Number of individuals (n), weight (g), size (mm) and sex of Procambarus clarkii captured at Villa Literno and Sessa Aurunca.

*Sites*	*n*	Mean Weight (g) ± SD	Mean Total Lenght (cm) ± SD	Sex
*Villa Literno (ViL)*	30	28.19 ± 4.43	9.58 ± 0.67	17 ♀ 13 ♂
*Sessa Aurunca (SeA)*	30	27.81 ± 3.51	9.32 ± 0.59	16 ♀ 14 ♂

**Table 2 animals-11-01933-t002:** Mean concentration (µg g^−1^ wet weigth) ± SEM of trace elements (As, Cu, Zn, Cr, Cd, Pb and Hg) in Procambarus clarki abdominal muscle (AbM) and hepatopancreas (Hep) from Villa Literno (ViL) and Sessa Aurunca (SeA).

Trace Elements (µg g^−1^ Wet Weight)	AbM ViL Site	Hep ViL Site	AbM SeA Site	Hep SeA Site
**As**	0.627 ^A^ ±0.173	8.534 ^B^ ±0.628	0.456 ^A^ ±0.092	3.248 ^B^ ±0.605
**Cu**	5.172 ^a^ ±0.450	8.577 ^b^ ±0.896	4.518 ^A^ ±0.461	11.512 ^A^ ±1.239
**Zn**	85.553 ^A^ ±1.788	216.643 ^B^ ±8.225	87.961 ±3.753	89.617 ±6.091
**Cr**	0.031 ^A^ ±0.002	1.265 ^B^ ±0.157	0.042 ^A^ ±0.016	1.328 ^B^ ±0.144
**Cd**	<dl	0.020 ±0.002	<dl	0.018 ±0.002
**Pb**	<dl	0.015 ±0.002	<dl	0.012 ±0.001
**Hg**	<dl	<dl	<dl	<dl

Probability levels for significant differences depending on organ type: AbM versus Hep: A, B: *p* < 0.01, a, b: *p* < 0.05.

## Data Availability

Detailed data supporting results are available on request at the corresponding author.

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
