# Peer review of "Heavy Metals in the Muscle and Hepatopancreas of Red Swamp Crayfish (Procambarus clarkii) in Campania (Italy)"

_animals, 2021, doi:10.3390/ani11071933_

Round 1

Reviewer 1 Report

The study deals with an interesting topic in the one-health context nowadays as regards heavy metals present in food.

The manuscript evaluates the occurrence of non-essential and essential elements in freshwater crayfish (Procambarus clarkii) edible tissues to establish the healthiness of this product and to evaluate the pollution status of the sampling sites. The topic is of interest for academics and for people because of the results obtained.

The article is well written and easy to understand by readers, the experimental design and the results are clearly exposed, statistical analysis is well performed, and the conclusions are supported by the obtained findings.

However, I suggest some modifications:

Abstract:

line 20: please remove “from”

Introduction:

line 56: please add a reference

lines 60-64: rewrite the sentence and add the reference

lines 86-87: avoid repetitions “selected for their potential high level of metal contamination”

line 89: please put P. clarkia in Italics (in all manuscript)

Materials and Methods

line 111: please change 0,5 ± 0,2 g in 0.5 ± 0.2 g

Results and discussion

The paragraphs contain both the full (hepatopancreas and abdominal muscle) and abbreviated (Hep and Abm) terms, please modify the text following one of the two ways.

Conclusion:

line 279: please, remove “the”

Author Response

Reviewer 1
We applied all editing the reviewer asked for.

Reviewer 2 Report

Review

Paper title: Heavy metals in muscle and hepatopancreas of red swamp crayfish (Procambarus clarkii) in Campania (Italy).

The authors analyzed edible tissues of red swamp crayfish for trace elements at 2 Italian sites. They found that the levels of heavy metals were lower than recommended by scientific organizations. These results are promising and provide a baseline for further studies.

All these reasons explain the relevance of the paper by Ariano Andrea and co-authors submitted to "Animals".

General scores.

The data presented by the authors are original and significant. All conclusions are justified and supported by the results. The study is correctly designed and technically sounds. In general, the statistical analyses are performed with good technical standards. We authors conducted careful work which will attract the attention of a wide range of specialists focused on the physiology of commercial shellfish and aquaculturists.

Main concerns.

L 95-96. More data about the two sampling sited are required (temperature regime, contamination levels, etc.).

Figure 1. The authors should include some localities (cities, towns) nearest to the sampling sites.

Table 1. “Weight range” and “Total length range” are incorrect in these cases because the authors presented mean values. They should include ranges (Min–Max) for both values. I also suggest to present the data separately for males and females. It would be useful to provide comparison details of the mean values between male and female P. clarkii (L. 167). The authors should explain why they pooled the data for males and females.

L 193-195, 196-199. The authors should explain these differences.

L 274. The authors concluded that “The higher Cu and Zn concentrations found in P. clarkii tissues, especially for Zn from ViL site, could be related to a higher anthropic activity in these areas”. This matter has not been dealt in the text. Please, clarify.

The authors should consider some recent papers:

Kuklina et al. (2013) Accumulation of heavy metals in crayfish and fish from selected Czech reservoirs. BioMed Research International. 2014, 306103

Xiong B (2020) Heavy metal accumulation and health risk assessment of crayfish collected from cultivated and uncultivated ponds in the Middle Reach of Yangtze River. Sci. Total. Env. 739, 139963

Mistri M. et al. (2020) Accumulation of trace metals in crayfish tissues: is Procambarus clarkii a vector of pollutants in Po Delta inland waters? Europ. Zool. J. 87, 46-57

Tan Y. et al. (2021) Human health risk assessment of toxic heavy metal and metalloid intake via consumption of red swamp crayfish (Procambarus clarkii) from rice-crayfish co-culture fields in China. Food Control 128, 108181

Specific comments.

L 2. Change “muscle” to “the muscle”

L 19-20. Change “included in the present study” to “from the Campania region (Italy)”

L 20. Delete “collected from”

L 22-27. Delete this section “Trace elements…human beings.”

L 31. Change “The research focused on P. clarkii,” to “P. clarkii is”

L 32. Change “crustacean” to “crustaceans”

L 33. Change “crayfish due to its trophic position, its way of feeding and its diet, can be considered as” to “the crayfish due to its trophic position and diet can be considered”

L 39. Change “In  conclusion,  data  obtained  from  this  study  showed” to “We suggest”

L 46. Change “scientific” to “the scientific”

L 55. Change “immune” to “the immune”

L 61. Change “metals” to “metal”

L 62. Change “ecosystem” to “ecosystems”

L 66. Delete “we included in the study”

L 68. Change “elements” to “element”

L 69. Change “indicator-species” to “indicator species”

L 71. Change “United” to “the United”

L 72. Change “purpose” to “purposes”

L 77. Delete “we  included  in  our  research,”

L 77. Change “in  geographic” to “on  geographic”

L 80. Change “1980’s” to “1980s”

L 81. Change “landfill” to “landfills”

L 83. Change “of the hinterland” to “the hinterland”

L 89. “P.  clarkii” should be italicized.

L 89. Change “bioindicator” to “a bioindicator”

L 96. Change “Campania” to “the Campania”

L 98. Change “weighted” to “weighed”

L 106. Change “P. clarkii” to “Procambarus clarkii

L 125. Delete “from mean”

L 127. Change “accumulation” to “concentrations”; “have both been” to “were”

L 128. Change “ANOVA” to “the ANOVA”

L 129. Change “elements” to “element”

L 130. Change “has been” to “was”

L 131. Change “elements” to “element”; “weigh” to “weight”

L 132. Change “normal” to “the normal”

L 133. Change “Significant value has been established at p<0.05” to “Significance level was set at p < 0.05.”

L 134. Change “has been” to “was”

L 140. Change “P. clarkii” to “Procambarus clarkii

L 142. Change “average concentration (µg g -1  wet weigth) ± SEM” to “standard errors”

L 144. Change “a variabilityof two trace elements concentration” to “variability in the concentration of two trace elements”

L 145. Change “levels” to “the levels”; “significatively” to “significantly”

L 153. Change “crayfish Hep than in AbM in” to “the crayfish Hep compared to AbM at”

L 156. Change “weigth” to “weight”

L 157. Change “P. clarkia” to “Procambarus clarkii

L 163. Change “crayfish” to “the crayfish”

L 168. Change “was no correlation” to “were no significant correlations”

L 175. Change “metals” to “metal”

L 177. Change “crayfish” to “the crayfish”

L 179. Change “muscle” to “the muscle”

L 180. Change “concentration” to “concentrations”

L 185. Change “elements” to “element”

L 187. Change “crayfish” to “the crayfish”

L 190. Change “crayfish” to “the crayfish”

L 193. Change “hepatopancreas” to “the hepatopancreas”

L 208. Change “hepatopancreas” to “the hepatopancreas”

L 211. Change “hepatopancreas” to “the hepatopancreas”

L 214. Change “what reported in literature” to “those reported in the literature”

L 216. Change “hepatopancreas” to “the hepatopancreas”

L 225. Change “governative” to “governance”

L 228. Change “it is essential” to “is essential”

L 232. Change “on data” to “in data”

L 241. Change “Italian” to “the Italian”

L 252. Change “in literature” to “in the literature”

L 253. Change “of a low risk” to “for low risk”

L 255. Change “mean concentrations detected in present study” to “the mean concentrations detected in the present study”

L 262. Change “muscle” to “the muscle”

L 265. Change “that the MRLs” to “than the MRLs”

L 267. Change “a good” to “good”

L 274. Change “a higher anthropic activity in these areas” to “higher anthropic activity in this area”

L 279. Change “the these” to “these”

Latin names of species should be italicized in the references.

L 308, 314, 317, 326, 331, 333, 335, 338, 345, 349, 352, 355, 358, 364, 376,

Author Response

Reviewer 2

Main concerns.

L 95-96. More data about the two sampling sited are required (temperature regime, contamination levels, etc.). We included a reference concerning trace elements in dogs from Sessa Aurunca, as there seems to be (to best of our knowledge) no information on pollution of the two areas.

Figure 1. The authors should include some localities (cities, towns) nearest to the sampling sites.

Table 1. “Weight range” and “Total length range” are incorrect in these cases because the authors presented mean values. They should include ranges (Min–Max) for both values. I also suggest to present the data separately for males and females. It would be useful to provide comparison details of the mean values between male and female P. clarkii (L. 167). The authors should explain why they pooled the data for males and females. Changed

L 193-195, 196-199. The authors should explain these differences. Done

L 274. The authors concluded that “The higher Cu and Zn concentrations found in P. clarkii tissues, especially for Zn from ViL site, could be related to a higher anthropic activity in these areas”. This matter has not been dealt in the text. Please, clarify. Done

The authors should consider some recent papers:

Kuklina et al. (2013) Accumulation of heavy metals in crayfish and fish from selected Czech reservoirs. BioMed Research International. 2014, 306103

Xiong B (2020) Heavy metal accumulation and health risk assessment of crayfish collected from cultivated and uncultivated ponds in the Middle Reach of Yangtze River. Sci. Total. Env. 739, 139963

Mistri M. et al. (2020) Accumulation of trace metals in crayfish tissues: is Procambarus clarkii a vector of pollutants in Po Delta inland waters? Europ. Zool. J. 87, 46-57

Tan Y. et al. (2021) Human health risk assessment of toxic heavy metal and metalloid intake via consumption of red swamp crayfish (Procambarus clarkii) from rice-crayfish co-culture fields in China. Food Control 128, 108181

  Included

Specific comments: all done
